# The Development of a Sensitive Droplet Digital Polymerase Chain Reaction Test for Quantitative Detection of Goose Astrovirus

**DOI:** 10.3390/v16050765

**Published:** 2024-05-11

**Authors:** Jianzhou Shi, Qianyue Jin, Xiaozhan Zhang, Jinbing Zhao, Na Li, Bingxue Dong, Jinran Yu, Lunguang Yao

**Affiliations:** 1The Shennong Laboratory, Zhengzhou 450046, China; shijian1109@163.com; 2School of Life Science, Nanyang Normal University, Nanyang 473061, China; 3Key Laboratory of Animal Immunology, Henan Academy of Agricultural Sciences, Zhengzhou 450002, China; 4College of Veterinary Medicine, Henan University of Animal Husbandry and Economy, Zhengzhou 450046, China; 5Henan Field Observation and Research Station of Headwork Wetland Ecosystem of the Central Route of South-to-North Water Diversion Project, Nanyang 473061, China

**Keywords:** goose astrovirus, droplet digital PCR, quantitative real-time PCR, ORF2 gene, detection

## Abstract

**Simple Summary:**

Goose astrovirus (GAstV) is a novel emerging pathogen that causes significant economic losses in waterfowl farming. In this study, we developed a droplet digital polymerase chain reaction (ddPCR) system for the sensitive and accurate quantification of GAstV using the conserved region of the ORF2 gene. The detection limit of ddPCR was 10 copies/µL, ~28 times greater sensitivity than quantitative real-time PCR (qPCR). The specificity of the test was determined by the failure of amplification of other avian viruses. The ddPCR test showed good repeatability and linearity, and the established ddPCR method had high sensitivity and good specificity to GAstV. Clinical sample test results showed that the positive rate of ddPCR was higher than that of qPCR. A convenient, sensitive, and specific detection method for GAstV in field samples is important in order to effectively control GAstV.

**Abstract:**

(1) Goose astrovirus (GAstV) is a novel emerging pathogen that causes significant economic losses in waterfowl farming. A convenient, sensitive, and specific detection method for GAstV in field samples is important in order to effectively control GAstV. Droplet digital polymerase chain reaction (ddPCR) is a novel, sensitive, good-precision, and absolute quantitation PCR technology which does not require calibration curves. (2) In this study, we developed a ddPCR system for the sensitive and accurate quantification of GAstV using the conserved region of the ORF2 gene. (3) The detection limit of ddPCR was 10 copies/µL, ~28 times greater sensitivity than quantitative real-time PCR (qPCR). The specificity of the test was determined by the failure of amplification of other avian viruses. Both ddPCR and qPCR tests showed good repeatability and linearity, and the established ddPCR method had high sensitivity and good specificity to GAstV. Clinical sample test results showed that the positive rate of ddPCR (88.89%) was higher than that of qPCR (58.33%). (4) As a result, our results suggest that the newly developed ddPCR method might offer improved analytical sensitivity and specificity in its GAstV measurements. The ddPCR could be widely applied in clinical tests for GAstV infections.

## 1. Introduction

Astroviruses (belonging to the *Astroviridae* family, genus Avastrovirus) are small, single-stranded, positive-sense, nonenveloped RNA viruses with a nonsegmented RNA genome approximately 6.0–7.7 kb in length; infection often leads to coinfections with many other enteric viruses, for instance, adenoviruses and noroviruses [1,2]. The astrovirus genome contains a 5′-untranslated region (UTR), a 3′-UTR, 3′ open reading frames (ORFs), and a poly(A) tail [3,4,5]. The genome is organized into 3 ORFs (ORF1a, ORF1b, and ORF2). Among them, ORF1a and ORF1b encode viral nonstructural proteins, and ORF2 encodes a viral structural protein (capsid) [1]. The host range of astroviruses is wide, and the main symptoms are diarrhea and intestinal disease [6,7,8]. Astrovirus infection poses a major problem for the poultry industry, resulting in many adverse effects, such as decreased egg production, reproductive disorders, insufficient weight gain, and even significantly increased mortality [9]. Goose astrovirus (GAstV) is a newly identified astrovirus which was first detected in China in November 2016 and which can cause gout and death in goslings. GAstVs are causative agents characterized by visceral urate deposition that cause deadly infection in 4- to 16-day-old goslings [3]. The deadly gout in geese caused by GAstV, the new virus, spread rapidly and caused severe economic losses to China’s goose industry in a short period of time [10,11,12,13].

Rapid, specific and sensitive accurate detection at the initial stage of virus infection is very important for effectively controlling the development of viral infectious diseases [14]. Currently, various assays, such as viral isolation, enzyme-linked immunosorbent assay (ELISA), and RT-qPCR, have been developed for the detection of GAstV. Nevertheless, these detection techniques and methods are either not highly sensitive, not sufficiently specific, or they cannot directly quantify viral nucleic acids, and are therefore not suitable for routine detection at the early stages of GAstV infection. While these methods have all played an important role in the detection of GAstV, more sensitive, specific, convenient, and reliable methods will provide richer and better detection resources for the detection of the virus. Hence, establishing a sensitive, rapid, simple, and reliable detection method is necessary for monitoring GAstV. The accurate detection and quantification of nucleic acids is a key step in diagnostics. Droplet digital polymerase chain reaction (ddPCR) is a novel technique that can accurately and absolutely quantify target nucleic acids in complex clinical samples. As a third-generation PCR technology, ddPCR based on droplets has the advantages of high sensitivity, high precision, and good repeatability. The ddPCR represents a new, more sensitive, more accurate, multiple-target gene quantification strategy for the detection of extremely small amounts of target nucleic acids [15]. The ddPCR has gained attention in preclinical studies as an accurate quantitative tool, especially for its highly accurate measurements [16]. The distinctive feature of ddPCR technology is that it can quantitatively detect the target nucleic acid in the sample without the need to establish a standard curve. This method has many advantages, such as a low inhibition rate, a high detection sensitivity, small changes in the quantitative limit, high analysis-accuracy, and high precision, but it reduces the dynamic range of detection [17]. The target-specific primers and fluorescent probes used in the ddPCR and TaqMan-based qPCR systems have been the same. The ddPCR consists of the following three steps: Step 1, the mixture of the reaction system is first dispersed into millions of water-in-oil droplets; Step 2, each droplet forms an independent small PCR system; Step 3, each droplet is classified as either positive or negative, depending on the difference in the detected fluorescence signal, and the proportion of positive droplets is counted. The target copy number in the sample is determined using Poisson’s algorithm [18,19]. In recent years, molecular technology has been widely used as a fundamental tool in veterinary diagnosis because it has shown stronger sensitivity, a more precise specificity, and greater rapidity in detecting viruses. In conclusion, as a third-generation absolute quantitative PCR technique, ddPCR technology has significant advantages, such as high sensitivity, high precision, and good repeatability [20]. However, there is, to date, no ddPCR method for detecting GAstV.

In this study, a new ddPCR strategy for the rapid, accurate, and sensitive diagnosis of GAstV, one which can not only be used for the detection of GAstV in goose clinical samples, but which also can accurately quantify the virus, has been established. In addition, the new ddPCR assay was compared with the qPCR assay in terms of sensitivity, specificity, stability, and repeatability.

## 2. Materials and Methods

### 2.1. Viruses and Clinical Samples

GAstV (GAstV XX strain, GenBank number: MN337323), FAdV-4 (fowl adenovirus), AIV-H9N2 (avian influenza), NDV (Newcastle disease virus), ALV (Avian leucosis virus), MDV-1 (Marek’s disease virus), MDV-CVI988, and MDV-3/HVT viruses were generously provided by the Henan Provincial Key Laboratory of Animal Immunology, Henan Academy of Agricultural Sciences (Zhengzhou, China). The clinical suspected samples were collected from the liver tissues of 36 goslings with gout from four goose farms in Henan province (Kaifeng, Xinxiang, Jiaozuo, and Shangqiu).

### 2.2. Design of Primers and Probes

The specific primers and probes were designed and synthesized based on the ORF2 gene of the AstV/SDPY/Goose/1116/17 strain (GenBank number: MH052598.1). The specific primers and probe were synthesized by Tsingke Biotech (Beijing, China) and modified (5′ FAM, 3′ BHQ1). The L1 sequence of the PCR upstream primer was 5′-GCCCAGATAGACAGCAGGAT-3′. The R1 sequence of the PCR downstream primer was 5′-GCGAGGGAGTAGCCTGTATT-3′. The sequence of probe P1 was FAM-ACCTGCCTCTGCCAGTGGCACC-BHQ1.

### 2.3. Nucleic Acid Extraction and Reverse Transcription

To optimize the ddPCR assay conditions, GAstV was used to collect leghorn male hepatocellular (LMH) cells infected with the GAstV XX strain. The infected LMH cells were extracted, the concentrations of RNA were quantified, and the viral nucleic acids were reverse transcribed. The rest of the collected virus and clinical samples were suspended in phosphate buffered saline (PBS, pH 7.4), and the virus or clinical samples were suspended in PBS at a ratio of 20%, *w*/*v*. The suspension was centrifuged at 4 °C at 8000 rpm for 15 min, after which the supernatant was retained. Viral nucleic acid and genomic DNA (FAdV-4, MDV-1, MDV-CVI988, and MDV-3/HVT) or RNA (GAstV, ALV, and AIV-H9N2) was extracted using the AxyPrep^TM^ Viral DNA/RNA Miniprep Kit (Axygen, Shanghai, China) according to the manufacturer’s instructions. Each viral nucleic acid RNA sample was reverse transcribed using the HiScript III All-in-one RT SuperMix reagent kit (Vazyme, Nanjing, China), following the product’s instructions. The cDNA/DNA was immediately amplified or stored at −80 °C for later use.

### 2.4. The ddPCR Assay

The viral copy numbers of GAstV and the clinical samples were quantified by a TD-1 Droplet Digital PCR System (TargetingOne, Beijing, China) according to the manufacturer’s instructions. Briefly, the 30 μL ddPCR mixture consisted of 7.5 µL of 4 × Unimix (TargetingOne, Beijing, China, final concentration: 1×), 2.4 µL of upstream primer (final concentration: 800 nM), 2.4 µL of downstream primer (final concentration: 800 nM), 0.75 µL of probe P1 (final concentration: 250 nM), and 1 µL (final concentration: <50 ng) of template, and diethyl pyrocarbonate (DEPC) DNase/RNase-free sterile water was added to a final volume of 30 µL. The 30 µL ddPCR system mixture and 180 µL oil were then loaded into the designated location of the droplet generation chip, and small droplets were generated on the droplet manufacturing machine.

PCR amplification was performed on a T100 thermal cycling apparatus (Bio-Rad Laboratories, Inc., Hercules, CA, USA). The amplification cycle conditions were as follows: 95 °C, 15 s–10 min, 1 cycle; 40 cycles of 94 °C for 5–30 s and 55–65 °C for 10–60 s; and then a final step at 12 °C for 5 min, 1 cycle, before storage at 4 °C. Moreover, the droplet temperature was increased at a rate of 1.5 °C/s in the T100 thermal cycler. After PCR amplification, a droplet chip reader (TargetingOne, Beijing, China) was used to read the PCR data for each small droplet unit; these were analyzed with droplet-reader software (TargetingOne, Beijing, China). The absolute value of the initial copy number of virus target nucleic acid molecules in each sample was accurately calculated according to Poisson statistics by considering the proportion of positive droplets relative to in the total number of droplets. To optimize the separation between positive droplets and negative droplets, the annealing temperature of the primer was optimized and analyzed (the temperature range was 54–61 °C). In other words, the optimal annealing temperature for ddPCR reactions was first determined by analyzing the PCR results at 54, 55, 56, 57, 58, 59, 60, and 61 °C. In addition, the primer-to-probe concentrations in the ddPCR system (800:166, 500:250, 600:166, and 400:166 nm) needed to be optimized to determine the optimal number of primers/probes. The ddPCR assay was repeated three times.

### 2.5. Limit of Blank (LoB) for ddPCR

Using two different batches of reagents, 4 negative samples (blanks) were tested for 3 consecutive days to obtain the value of LoB.
*Cp* = 1.645 × [1 − (4B − 4K)^−1^]^−1^
LoB = M + *Cp*SD

*Cp* represents the multiplier of the 95th percentile of the normal distribution (α = 0.05). B is the number of blank samples during the actual test. K is the number of blank samples. M is the mean of the blank. SD is the standard deviation of the blank.

### 2.6. QPCR Assay

In this study, the primers and probes used for qPCR were the same as those used for ddPCR. The qPCR test was performed on the GAstV and samples using a Bio-Rad C1000 Touch^TM^ Thermal Cycler. The 30 µL final reaction system consisted of 15 µL of 2 × Mix (AceQ Universal U+ Probe Master Mix V2, Vazyme, Nanjing, China), 0.8 µL (10 µM) of each of the reverse and forward primers, 0.5 µL (10 µM) of probe, 11.9 µL of RNase-free ddH_2_O, and 1 µL of template. Then, qPCR amplification was performed at 37 °C for 2 min, 95 °C for 5 min, 45 cycles at 95 °C for 10 s, and at 59 °C for 30 s. After the qPCR, a standard curve was drawn. Specificity, sensitivity, and repeatability were tested by qPCR.

### 2.7. Sensitivity Test of ddPCR and qPCR

First, the nucleic acid of GAstV was extracted and reverse-transcribed into cDNA, and the DNA concentration was obtained by continuous dilution to compare the sensitivity and accuracy of the two amplification systems, ddPCR and qPCR. The theoretical copy numbers for virus dilution were, in order, 200,000, 20,000, 2000, 200, 20, 10, and 2. For the purpose of comparison, qPCR and ddPCR tests were performed separately while using the same template. Eight replicates were detected for each concentration. DEPC sterile water was used as a blank control for 8 replicates. The correlation of the qPCR and ddPCR standard curves was used to analyze and evaluate the sensitivity and quantitative consistency of the measured results of the two detection methods.

### 2.8. Specificity and Reproducibility of ddPCR

To evaluate the specificity of the ddPCR method, nucleic acids from GAstV and other seven other common avian viruses (FAdV-4, H9N2, NDV, ALV, MDV-1, MDV-CVI988, and MDV-3/HVT) were used as reaction templates and tested with a GAstV-specific primer and probe for ddPCR. For the negative control, we used nuclease-free ddH_2_O as a template. Specificity testing was performed using optimized conditions. The established ddPCR method was used to detect GAstV at different copy numbers to evaluate the sensitivity, robustness, and repeatability of the ddPCR detection. Each sample was analyzed in triplicate to assess inter-assay and intra-assay repeatability.

### 2.9. Clinical Sample Detection by ddPCR and qPCR Assays

A total of 36 clinically suspected samples were tested using the ddPCR and qPCR in order to assess the sensitivity of the established ddPCR assay. For each round of ddPCR and qPCR reaction, one positive control and one negative control were used simultaneously. The conditions for qPCR amplification were as described above, and ddPCR was performed under the optimized amplification conditions. The positive detection rates of the two methods, ddPCR and qPCR, were calculated, and the sensitivities of the two detection methods were compared. To evaluate the consistency of quantification, quantitative values were determined for each sample by both ddPCR and qPCR methods.

### 2.10. Statistical Analysis

In this study, all the data statistical analysis and data plotting were performed using GraphPad Prism software (version 5.0; La Jolla, CA, USA).

## 3. Results

### 3.1. Development of a GAstV ddPCR Assay

To optimize the annealing temperature of ddPCR, the temperature gradients from 54 to 61 °C (at the following temperatures: 54, 55, 56, 57, 58, 59, 60, and 61 °C) were used in the ddPCR assay. Figure 1 shows that the difference between the positive and negative droplet fluorescence signals was the greatest at 59 °C, which resulted in the largest number of positive droplets, that is, the largest number of amplified products, and the largest difference in fluorescence amplitude between the positive (blue fluorescence signals) and negative controls (grey fluorescence signals). Therefore, 59 °C was chosen as the optimal annealing temperature for the ddPCR reaction procedure. Then, the amounts of primers and probe in the ddPCR system were optimized. As shown in Figure 2, the results indicated that the optimal concentration ratio of primer–probe in the ddPCR system was 800:166 nM, because this ratio resulted in optimal separation of positive (blue) and negative (gray) droplets. Thus, the reaction conditions set for optimizing the GAstV ddPCR assay were 59 °C as the optimum annealing temperature of the reaction system and a primer-to-probe concentration of 800:166 nM as the optimal input amount.

### 3.2. Limit of Blank (LoB) for ddPCR

To establish the LoB, we analyzed 4 blank samples with two different batches of reagents for 3 consecutive days (Table 1). The LoB was estimated nonparametrically as the 95th percentile of the measurements. The LoB of reagent batch 1 was 1.78, and the LoB of reagent batch 2 was 1.81. Both of these were rounded to the whole number 2. That is, the LoB of this method was 2. The LoB of the capsid protein-encoding gene (ORF2) gene was calculated to be two copies/reaction. We conducted further evaluation using four known negative samples and confirmed that the highest copy/reaction number was 2. Therefore, based on this, we set the cut-off for ddPCR results for values reported as “undetectable” at less than two positive copies/reaction.

### 3.3. Analytical Sensitivity and Reproducibility

The serially diluted cDNA of GAstV showed a good linear relationship in both qPCR and ddPCR detections. In the ddPCR assay, the standard curve had a good linear correlation (Y = 1.0241X − 0.0991), with an R^2^ value of 0.9984 (Figure 3a). By comparison, the standard curve for qPCR detection was Y = 0.9599X + 0.3954, with an R^2^ value of 0.9593 (Figure 3b). As shown in Table 2, the test determined that 10 copies/µL was the detection limit of the ddPCR assay. In contrast, the virus detection limit of the qPCR assay was 280 copies/µL. With a cut-off detection limit of 45 cycles, the detection limit of the ddPCR assay was 28 times lower than that of the qPCR assay.

In the reproducibility tests of the ddPCR assay, the measured intra-assay coefficient of variation (CV) ranged from 0.35% to 4.34%, and the measured CV of the inter-assay ranged from 0.36% to 4.45% (Table 3). In the repeatability tests for the qPCR assay, the measured intra-assay CV ranged from 1.98% to 17.91%, and the measured inter-assay CV ranged from 0.72% to 18.04% (Table 4). The results showed that the established ddPCR method for detecting GAstV has good repeatability.

### 3.4. Analytical Specificity of the ddPCR Assay

For the specificity of ddPCR analysis, DNA/RNA nucleic acid templates of different poultry viruses, including GAstV, FAdV-4, H9N2, NDV, ALV, MDV-1, MDV-CVI988, and MDV-3/HVT, were prepared. As shown in Figure 4, only the GAstV test was positive (14,375 copies/µL), while the tests used for the detection of other viruses were negative (0 copies/µL). The results showed that the established ddPCR assay exhibited good specificity for detecting GAstV.

### 3.5. Clinical Sample Testing

To further determine the clinical practicality of the ddPCR assay, the ddPCR assay and qPCR assay were evaluated on 36 clinical samples collected from goose farms in Henan Province, China. As shown in Table 5, GAstV was detected with a positive rate of 88.89% (32 of 36) by ddPCR and 58.33% (21 of 36) by qPCR. Among these results, 11 samples had inconsistent detection results: the qPCR results were negative, and the ddPCR results were positive. To rule out false-positives, the 11 samples were retested three times by ddPCR assay. In the presence of effective NTC (negative control, ddH_2_O), the ddPCR results of these 11 samples were all positive. Based on the data of GAstV detection data from clinical samples, the sensitivity of the ddPCR assay was superior to that of the qPCR assay.

## 4. Discussion

Astroviruses are small, nonencapsulated, single-stranded positive RNA viruses belonging to the *Astroviridae* family. The rate of infection is as high as 80%, and the mortality rate is approximately 50% [11,21,22]. The possibility of cross-species transmission of GAstV from infected geese [23] to ducks [24,25] and chickens [26] has recently been reported [14]. Continuous monitoring of GAstV is essential for effective control and prevention; therefore, an effective and rapid diagnostic method is needed in order to monitor and detect the spread of GAstV.

At present, the following detection methods have been reported: real-time reverse transcription polymerase chain reaction (RT-PCR) [27,28,29,30], TaqMan-probe-based real-time RT-qPCR [31,32], peptide-based ELISA [33], indirect competitive ELISA [34], one-step reverse transcription loop-mediated isothermal amplification [35], reverse transcription-enzymatic recombinase amplification coupled with a CRISPR-Cas12a system [14], and immunochromatographic strip assay [36]. In the past few years, ddPCR has undergone rapid development and has been widely applied for the detection and quantitative analysis of a variety of viruses [37,38]. Today, it has become a promising tool for virus detection [18,39].

The ddPCR assay has been shown to have greater diagnostic sensitivity and specificity than the qPCR assay, particularly because very small amounts of nucleic acid can be used normally [18,19]. The next-generation technology of ddPCR can achieve the most absolute quantification by partitioning the reaction. This highly accurate and highly sensitive molecular detection technology is widely used in biomedical fields such as extremely tiny DNA detection, rare gene mutation detection, and copy number variation detection, etc. [16,40,41].

In this study, a new ddPCR method for the detection and quantification of GAstV was developed. The method has the significant advantages of high sensitivity, good specificity, and low intra-assay and inter-assay coefficients of variation (<4.45%), indicating that this method can provide rapid, accurate, convenient, and repeatable results for the diagnosis of GAstV infection in animals. This study is the first to use ddPCR for the detection of GAstV. After optimizing the ddPCR procedure (the annealing temperature) and reaction procedure (primer-to-probe concentration), the detection limit of the ddPCR assay was 10 copies/rection, which was 28 times greater than that of the qPCR assay. The designed primers and probes detected most of the GAstV strains of the avian astrovirus Group 1. In addition, the detection limits of TaqMan real-time RT-PCR [29], qRT-PCR [31], SYBR Green I real-time PCR [42], qPCR [32], and immunochromatographic strips (ICS) [36] were 33.3 copies/µL, 33.4 copies/µL, 6.58 × 101 copies/μL, 100 copies/µL, and 1.2 μg/mL, respectively, which demonstrated lower sensitivity than the ddPCR assay. Especially when the sample to be tested contains low levels of viral nucleic acid, higher sensitivity and repeatability can help improve the reliability of positive detection rates in clinical samples.

The established ddPCR method was used to detect 36 clinically suspected samples, and the detection results were evaluated. The results showed that 11 samples were positive for ddPCR and negative for qPCR, indicating that ddPCR detection indeed has a higher GAstV detection rate and sensitivity than qPCR assay. The LoB value is an important parameter for determining the detection ability of ddPCR technology. For samples that did not contain analytes (blank samples), LoB had the highest number of false-positives, representing the best detection results as to false-positives. Multiple blank samples could be tested over multiple days and multiple times, and all tests could be statistically analyzed with 95% confidence. In simple terms, LoB defined the standard for 0 concentration samples. The LoB was calculated to be 2 copies/reaction. As a result, false-positive results were not detected in clinical samples using the established ddPCR detection method. The identification of LoB could exclude the possibility of false-positives when using this ddPCR method. This result indicated that ddPCR was more suitable for the early diagnosis of GAstV infection and might help to monitor GAstV to better prevent and control its epidemic spread. Another outstanding advantage of the ddPCR method is that absolute quantification can be achieved without establishing a standard curve. In contrast, the qPCR method can only be used for quantitative detection if the calibration curve generated by the template is based on continuous dilutions, and the calculation of the copy number depends on the Ct value of the standard curve. The ddPCR method is more convenient than the qPCR method because no standard curve is needed. In addition, the specificity of the ddPCR method was good, and the results of avian disease virus nucleic acid detection for FAdV-4, AIV-H9N2, NDV, ALV, MDV-1, MDV-CVI988, and MDV-3/HVT were negative. Therefore, the ddPCR method established in this study provides a specific, sensitive, convenient, and reliable new method for the detection and quantification of viruses.

The upper limit of the concentration of DNA samples detected by this method was 2 × 10^5^ copies/µL. If the concentration surpasses this threshold, ddPCR cannot be used for quantification, and the sample must be diluted. This conclusion is consistent with previous reports [18].

## 5. Conclusions

Overall, in this study, a sensitive, specific, reliable, and convenient ddPCR method for detecting GAstV was established and evaluated in clinical samples. Compared with the traditional qPCR method, the established ddPCR method had higher sensitivity, stronger specificity, and better repeatability, which is conducive to the clinical detection and epidemiological investigation of GAstV.

## Figures and Tables

**Figure 1 viruses-16-00765-f001:**
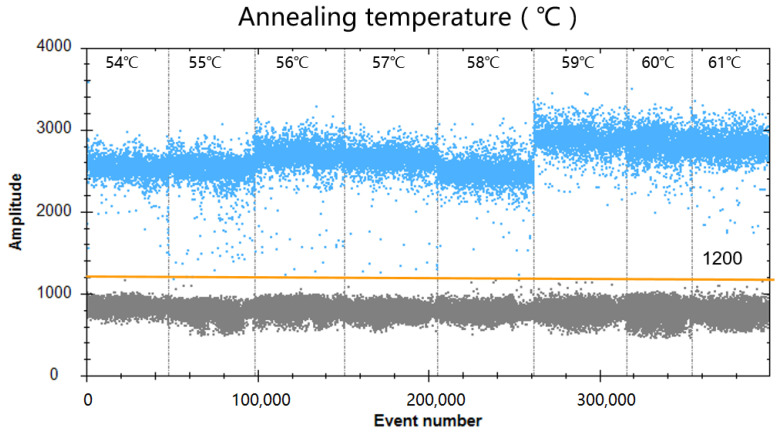
Effect of annealing temperature on GAstV ddPCR system. The assay was run under annealing temperature gradients of 54, 55, 56, 57, 58, 59, 60, and 61 °C, respectively.

**Figure 2 viruses-16-00765-f002:**
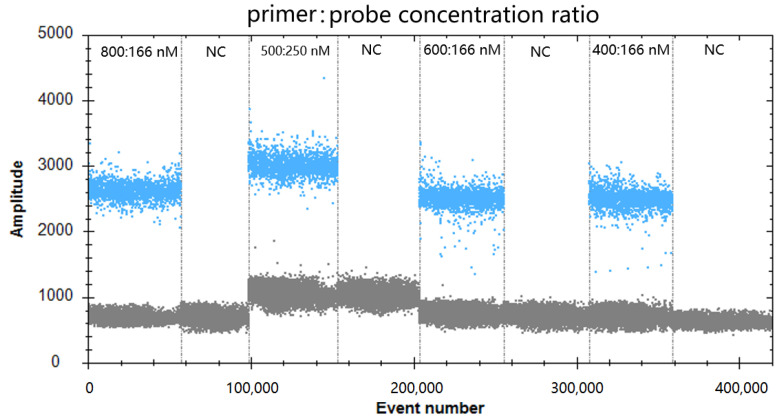
Effect of primer-to-probe concentration ratio on the GAstV ddPCR system. The assay was conducted across a primer and probe concentration ratio gradient: 800:166, 500:250, 600:166, and 400:166. NC, no template control.

**Figure 3 viruses-16-00765-f003:**
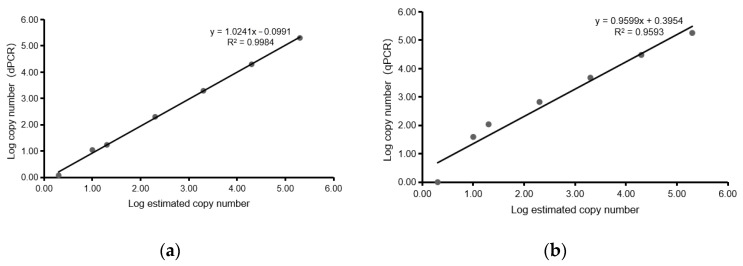
Quantification of serially diluted cDNA of GAstV by ddPCR and qPCR. (**a**) Standard curves for cDNA of GAstV, constructed by ddPCR. The quantification correlation was obtained by plotting the log assumed concentration against the log starting concentration. (**b**) Standard curves for cDNA of GAstV, constructed by qPCR. The quantification correlation was obtained by plotting the log assumed concentration against the log starting concentration.

**Figure 4 viruses-16-00765-f004:**
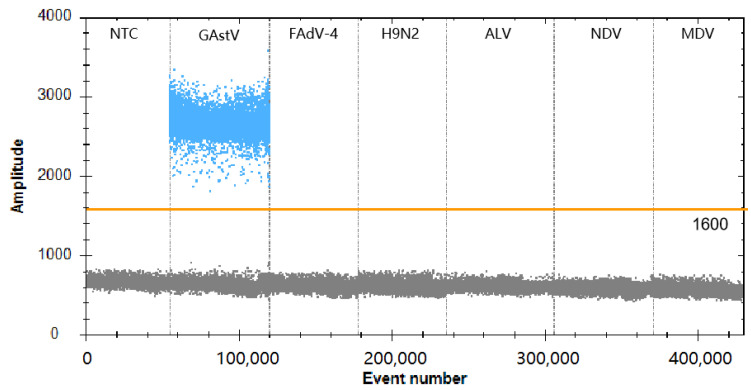
Specificity analysis of the GAstV ddPCR assay. Lanes 1–7 (divided by vertical black dotted lines): the fluorescence amplitudes of NTC (negative control, ddH_2_O), GAstV, FAdV-4, H9N2, ALV, NDV, and MDV, respectively.

**Table 1 viruses-16-00765-t001:** Limit of blank (LoB) value determination of droplet digital PCR (ddPCR).

Reagent	Test Days	Number of Tests	Sample 1	Sample 2	Sample 3	Sample 4
Reagent batch 1	1	1	0	0	0	1.1
		2	1.1	1.1	1.1	0
		3	0	0	0	0
	2	1	0	0	0	0
		2	2.3	1.2	1.3	1.2
		3	0	1.4	0	0
	3	1	0	0	0	1.1
		2	0	1.3	0	0
		3	2.6	0	2.3	0
Reagent batch 2	1	1	0	0	0	0
		2	1.2	2.7	5.8	0
		3	2.2	4.7	0	1.2
	2	1	0	0	1.1	0
		2	0	1.2	0	0
		3	0	0	0	1.2
	3	1	0	2.2	1.2	3.8
		2	2.3	0	0	0
		3	0	0	0	0

**Table 2 viruses-16-00765-t002:** Detection limits of quantitative real-time PCR (qPCR) and droplet digital PCR (ddPCR).

Input of GAstV RNA Copy Number	qPCR Hit Rate (Positive/Total)	ddPCR Hit Rate (Positive/Total)
500	1.00 (36/36)	ND
300	0.97 (35/36)	ND
200	0.94 (34/36)	ND
100	0.31 (11/36)	ND
50	ND	1.00 (36/36)
20	ND	1.00 (36/36)
10	ND	0.97 (35/36)
5	ND	0.64 (23/36)
2	ND	0.25 (9/36)
1	ND	0.08 (3/36)
NTC	0.00 (0/24)	0.00 (0/236)
LoD	280	10

ND, not detected; NTC, negative control (ddH_2_O); LoD, limit of detection.

**Table 3 viruses-16-00765-t003:** Robustness and reproducibility analysis of droplet digital PCR (ddPCR).

Concentration of GAstV RNA (copies/µL)	Intra-Assay Variation (Robustness)			Inter-Assay Variation (Reproducibility)		
	Mean of Detected Concentration (copies/µL)	SD	CV (%)	Mean of Detected Concentration (copies/µL)	SD	CV (%)
200,000	197,745.6	558.1	0.35	199,579.1	584.7	0.36
20,000	19,550.5	399.9	2.50	19,625.1	278.0	1.74
2000	1959.4	32.5	2.03	1938.0	32.4	2.05
200	205.0	3.9	2.32	196.6	5.1	3.19
20	19.7	0.7	4.34	19.3	0.7	4.43
10	9.8	0.3	4.08	10.3	0.4	4.45

CV, coefficient of variation; SD, standard deviation.

**Table 4 viruses-16-00765-t004:** Robustness and reproducibility analysis of quantitative real-time PCR (qPCR).

Concentration of GAstV RNA (copies/µL)	Intra-Assay Variation (Robustness)			Inter-Assay Variation (Reproducibility)		
	Mean of Detected Concentration (copies/µL)	SD	CV (%)	Mean of Detected Concentration (copies/µL)	SD	CV (%)
200,000	171,717.4	5040.0	2.94	179,935.8	1303.1	0.72
20,000	27,794.6	549.4	1.98	30,952.9	1012.0	3.27
2000	2462.0	126.0	5.12	2668.7	481.6	18.04
500	685.3	45.7	6.67	674.1	55.4	8.21
200	73.9	13.2	17.91	58.9	9.1	15.41

CV, coefficient of variation; SD, standard deviation.

**Table 5 viruses-16-00765-t005:** Comparison of ddPCR and qPCR sensitivity for GAstV clinical samples.

		ddPCR		Total
		Positive	Negative	
qPCR	Positive	21	0	21
	Negative	11	4	15
Total		32	4	36

## Data Availability

The original contributions presented in the study are included in the article, further inquiries can be directed to the corresponding author.

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
