# Peer review of "The Development of a Sensitive Droplet Digital Polymerase Chain Reaction Test for Quantitative Detection of Goose Astrovirus"

_viruses, 2024, doi:10.3390/v16050765_

Round 1

Reviewer 1 Report

Comments and Suggestions for Authors

In this manuscript, Shi et al. develop and validate a droplet digital PCR for the quantitative detection of goose astrovirus (later used on some clinical samples), including determining optimal annealing temperature, primer-probe ratio, and limit of detection. Additionally, the study determines specificity after performing the assay on a variety of DNA and RNA viruses, but not other avian astroviruses.

Comments:

Simple summary-this seems to be an addition based on the template and instructions and is up to the editor if this should be included.

Lines 183-184: How were the theoretical copy numbers determined?

To determine Goose astrovirus specificity, the assay needs to be run on related avian astroviruses (duck, chicken, turkey).

This reviewer has seen recommendations for quantitative PCRs to include competition assays. This includes keeping the target nucleic acid the same and add increasing amounts of a different nucleic acid (and vice versa). To this reviewer that seems unnecessary and I am not requesting it, but if another reviewer does I will be ok with it and review the data.

Author Response

Review Report 1

Thank you very much for taking the time to review this manuscript.

  1. Simple summary-this seems to be an addition based on the template and instructions and is up to the editor if this should be included.

Answer:

The paper template includes a "Simple summary."

  1. Lines 183-184: How were the theoretical copy numbers determined?

Answer:

Lines 183-184:  Sensitivity test for ddPCR and qPCR

First, the nucleic acid of GAstV was extracted and reverse-transcribed into cDNA, and the DNA concentration was obtained by continuous dilution to compare the sensitivity and accuracy of the two amplification systems, ddPCR and qPCR. The theoretical copy numbers for virus dilution were, in order, 200000, 20000, 2000, 200, 20, 10, and 2.”

The ddPCR method can be quantified, and the collected GAstV is detected and quantified in the pre-experiment. Once we've determined the initial concentration, we can dilute it in turn.

  1. To determine Goose astrovirus specificity, the assay needs to be run on related avian astroviruses (duck, chicken, turkey).

Answer:

This ddPCR method for GAstV detection was established using GAstV XX strain  (GenBank number: MN337323).  The specific primers and probes in this ddPCR assay were designed based on of avian astrovirus Group 1 (GenBank number: MH052598.1).  The specific primers and probes were designed and synthesized based on the ORF2 gene of AstV/SDPY/Goose/1116/17 strain (GenBank number: MH052598.1).

During the isolation and identification of goose astrovirus XX strain (GenBank login number: MN337323), our team has conducted homology analysis and other correlation analysis on the genome, ORF1b and ORF2 genes with representative a large number of strains. The experimental results have been reported.

Genome:

The nucleotide homology of GAstV XX strain and representative strains GD, AstV/SDPY/Goose/1116/17 and AAstV/Goose/CHN/2017/SD01 causing gout in goslings was 98.1%, 98.7% and 98.7%, respectively.

The gene of ORF1b、ORF2:

The nucleotide homology of ORF1b and ORF2 genes of GAstV XX strain with GD, AstV/SDPY/ Goose/1116/17 and AAstV/Goose/CHN/2017/SD01, which were representative strains causing gout in goslings, was 98.6% ~ 99.4%, The homology of amino acid ranged from 98.6% to 99.8%.

Phylogenetic analysis results showed that GAstV XX was in the same evolutionary clade as GD, AstV/SDPY/Goose/1116/17, AAstV/Goose/CHN/2017/SD01. They all belong to avian astrovirus Group 1 [1]. It can be seen that the homogeneity of AstV/SDPY/Goose/1116/17 strain was higher than that of the former viral strain causing the wind pain in goslings. AstV/SDPY/Goose/1116/17 strain is a representative strain of avian astrovirus Group 1. The designed primers and probes can detect most GAstV of avian astrovirus Group 1.

[1] Qianyue Jin, Yonggang Guo, Junpeng Li, et al. Isolation, identification and genetic characterization of goose astrovirus XX strain[J]. Journal of Henan Agricultural Sciences.2021,50(6):134-141. doi:10.15933/ j.cnki.1004-3268.2021.06.016

In short, the isolated GAstV XX strain (GenBank number: MN337323) was sequenced and its genetic characteristics were analyzed. Phylogenetic analysis showed that GAstV XX strain was in the same evolutionary branch as the current circulating strain causing gout in goslings, belonging to group 1 of avian astroviruses, and had a distant evolutionary relationship with duck, Turkey and chicken astroviruses, with great differences. The amino acid sequence analysis of ORF2 encoded protein of different geese strains showed that there was little difference among the circulating strains, and only some mutations of amino acid sites existed.

 Our team is collecting the disease materials of avian astroviruses (duck, chicken, turkey) and will supplement the astroviruses from other birds in future experiments.

  1. This reviewer has seen recommendations for quantitative PCRs to include competition assays. This includes keeping the target nucleic acid the same and add increasing amounts of a different nucleic acid (and vice versa). To this reviewer that seems unnecessary and I am not requesting it, but if another reviewer does I will be ok with it and review the data.

Answer:

The objective was to compare the sensitivity of ddPCR and qPCR detection methods.

Reviewer 2 Report

Comments and Suggestions for Authors

Here some major concerns should be clarified.

1.     Figure 3, there must be something wrong, Figure 3-A and Figure 3-B are the same?

2.     Now they are two different GAstV genotype in China, we suggested two GAstV genotype should be add and then compared the results.

3.     The 3.5. Clinical sample testing section, the copy number of ddPCR and qPCR should be added especially for the different data. Also, the positive rate of ddPCR and qPCR confusing us. More details should be discussed at discussion section.

Author Response

Review Report 2

Thank you very much for taking the time to review this manuscript

  1. Figure 3, there must be something wrong, Figure 3-A and Figure 3-Bare the same?

Answer:

     The Figure 3-B has been revised.

Figure 3.  (a) Standard curves cDNA of GAstV constructed by ddPCR.  

(b) Standard curves of cDNA of GAstV constructed by qPCR.

  1. Now they are two different GAstV genotype in China, we suggested two GAstV genotype should be add and then compared the results.

Answer:

     The designed primers and probes can detect most GAstV of avian astrovirus Group 1.

In this study, to evaluate the specificity of the ddPCR method, nucleic acids from GAstV and other seven other common avian viruses (FAdV-4, H9N2, NDV, ALV, MDV-1, MDV-CVI988, and MDV-3/HVT) were used as reaction templates and tested with a GAstV specific primer and probe for ddPCR.

Our team is collecting the disease materials of avian astroviruses Group 2 (duck, chicken, turkey) et al., and will supplement the astroviruses from other birds in future experiments.

  1. The 3.5. Clinical sample testing section, the copy number of ddPCR and qPCR should be added especially for the different data.

Answer:

When each clinical sample is tested, first of all, the initial amount of virus is not quantified and uniform; Second, when testing clinical samples, as long as the virus is detected under Limit of blank (LoB) conditions, it is judged to be positive, and conversely, the virus is not detected, it is judged to be negative. Therefore, it is not necessary to supplement the copy number of ddPCR and qPCR.

  1. Also, the positive rate of ddPCR and qPCR confusing us. More details should be discussed at discussion section.

Answer:

The Limit of blank (LoB) of ddPCR was added and the results were obtained.

① Limit of blank (LoB) of ddPCR
Using two different batches of reagents, 4 negative samples (blank) were repeated for 3 consecutive days to obtain the value of LoB.

Cp=1.645*[1-(4B-4K)-1] -1 

LoB =M + CpSD

Cp represents the multiplier of the 95th percentile of the normal distribution (α=0.05). B is the number of blank samples during the actual test. K is the number of blank samples. M is mean of blank. SD is standard deviation of blank.

To establish the LoB, we analyzed 4 blank samples with two different batches of reagents for 3 consecutive days. The LoB was estimated nonparametrically as the 95th percentile of the measurements. The LoB of reagent batch 1 was 1.78, and the LoB of reagent batch 2 was 1.81. Both of these were rounded to the whole number 2. That is, the LoB of this method was 2. The LoB of the capsid protein gene (ORF2) gene was calculated to be 2 copies/reaction. We further evaluated using 4 known negative samples and confirmed that the highest copy/reaction number was 2. Therefore, based on this, we set the cut-off for ddPCR results for values reporting as “undetectable” at less than 2 positive copies/reaction.

LoB (Limit of Blank) is an important parameter to judge the detection ability of ddPCR technology. For samples that do not contain analytes (blank samples), LoB is the highest detection result that can be observed. In other words, the maximum number of false positives. The LoB is necessary to determine whether the reaction is positive or negative, and to calculate the LoD (Limit of Detection). In ddPCR, due to the influence of various factors such as reagent components, experimental environment, instruments, etc., even if the sample does not contain the substance to be tested, sometimes the result may not be 0 copy, causing false positives, so it is necessary to define these results that do not have clinical significance, which is LoB. That is, the highest measurement that can be observed in a blank sample.

LoB means that the vast majority (95%) of the zero concentration samples tested are negative and cannot be false positive.

For example, the LoB is 30 copies/mL, although the test result is 10 copies/mL, it is understood that the sample does not contain the substance to be measured, or the concentration is too low to be detected at all. Multiple blank samples can be tested for multiple days and multiple times, and all tests can be statistically analyzed with 95% confidence. In simple terms, LoB defines the standard for 0 concentration samples.

The following results were obtained by supplementing the LoB experiment results. The LoB was calculated to be 2 copies/reaction. We further evaluated using 4 known negative samples and confirmed that the highest copy/reaction number was 2. Therefore, based on this, we set the cut-off for ddPCR results for values reporting as “undetectable” at less than 2 positive copies/reaction.

In summary, false positive results would not be found in clinical samples using the established ddPCR detection method. The identification of LoB can exclude the possibility of false positive in the ddPCR method.

② In the study, the detection limit of ddPCR assay was 10 copies/rection, the virus detection limit of the qPCR assay was 280 copies/µL, which was 28 times higher than that of qPCR detection.The detection rate of ddPCR was indeed higher than that of qPCR in clinical tests with low virus content.

③ The 11 qPCR-negative suspected samples were detected by conventional PCR (amplification with primers of ORF2), and no target bands were amplified.

④ All samples identified positive by qPCR were confirmed by ddPCR.

⑤ PCR amplification products tested positive using ddPCR were encapsulated in oil droplets and cannot be recovered or sequenced.

Reviewer 3 Report

Comments and Suggestions for Authors

The manuscript by Shi et al. described the development of a novel, sensitive, specific and accurate method to detect Goos Astrovirus gene with good repeatability by using droplet digital PCR. It is well understood that ddPCR is superior to qRT-PCR, one of the conventional methods. However, I think there are some improvements that can be made in the way the results are presented, and I would like you to consider them.

Materials and Methods

1. The contents cited as Table 1 in line 116 do not match the Table 1 in this manuscript.

2. Since the verb cannot be found, the meaning of the sentence in lines 123-125 cannot be understood.

3. Should “TM” on lines 130 and 173 be superscripted?

4. For "2.4. ddPCR assay" and "2.6. QPCR assay", are the criteria for positive and negative results provided?

5. How long does ddPCR and qPCR each take to carry out? If the comparison of time required for these tests is valid, how about comparing them? 

Result

6. What do the orange bars in Figure 1 and 4 mean?

7. I think Table 1 in the manuscript is related to "3.2 Limit of blank (LoB) of ddPCR", but there is no citation in the text.

8. Figure 3(b) is the same graph as (a).

9. Why are the detection limits 10 (ddPCR) and 280 (qPCR)? The reason why the authors decided are not explained.

10. What is “LoD” in Table 2?

11. Does "qPCR" in the title of Table 4 also need the full name?

12. Why were the results of 4 samples negative in ddPCR? As pointed out in Materials and Methods, are there any criteria negative for ddPCR?

Discussion

13. At lines 443~444, the author says this ddPCR could detect most GAstV. GAstV XX strain was used in the present study (L123). What is the basis that other strains can be detected by the ddPCR?

14. Were true positive or true negative of 36 clinical samples confirmed by viral isolation test? 

15. Is it possible to compare the ddPCR results (copy number) and the qRT-PCR results (copy number) for each of the 36 specimens using a distribution chart or scatter diagram? Because the positive/negative results in Table 5 are not enough to correctly understand the usefulness of ddPCR. Similarly, from the information in Table 5 alone, it is difficult to know whether it can be said that there were no false positives in ddPCR. 

Comments on the Quality of English Language

There were some sentences that appeared to have incorrect grammar.

Author Response

Review Report 3

Thank you very much for taking the time to review this manuscript

Materials and Methods

  1. The contents cited as Table 1 in line 116 do not match the Table 1 in this manuscript.

Answer:

“(Table 1)” in line 116 has been deleted. The full tables and figures were also checked one by one.

  1. Since the verb cannot be found, the meaning of the sentence in lines 123-125 cannot be understood.

Answer:

Original: “The infected LMH cells for the extraction, quantification of the concentrations of RNA and reverse transcription of viral nucleic acids.”

Revision: “The infected LMH cells were extracted, the concentrations of RNA were quantified and the viral nucleic acids were reverse transcribed.” 

  1. Should “TM” on lines 130 and 173 be superscripted?

 Answer:

The "TM" of lines 130 and 173 have been modified to superscript.

  1. For "2.4. ddPCR assay" and "2.6. QPCR assay", are the criteria for positive and negative results provided?

Answer:

Each time on the machine test, positive control and negative control will be set at the same time. There are positive and negative comparisons between these two parts of "2.4. ddPCR assay" and "2.6. QPCR assay". The positive control was GAstV (GAstV XX strain), and the negative control was NTC (no template control), NC (negative control, ddH2O).

  1. How long does ddPCRand qPCR each take to carry out?  If the comparison of time required for these tests is valid, how about comparing them? 

 Answer:

The time required for ddPCR and qPCR detection is similar, both are PCR processes, and the time required for ddPCR droplets preparation and droplets identification is about 5 min for each sample. The difference between the two is mainly in terms of sensitivity, ddPCR can detect 2 copies/reaction.

Result

  1. What do the orange bars in Figure 1 and 4 mean?

 Answer:

Partition threshold

  1. I think Table 1 in the manuscript is related to "3.2 Limit of blank (LoB) of ddPCR", but there is no citation in the text.

Answer:

"3.2 Limit of blank (LoB) of ddPCR" has been revised.

To establish the LoB, we analyzed 4 blank samples with two different batches of reagents for 3 consecutive days (Table 1).

  1. Figure 3(b) is the same graph as (a).

Answer:

The Figure 3-B has been revised.

Figure 3.   (a) Standard curves cDNA of GAstV constructed by ddPCR.  

(b) Standard curves of cDNA of GAstV constructed by qPCR.

  1. Why are the detection limits 10 (ddPCR) and 280 (qPCR)? The reason why the authors decided are not explained.

Answer:

It is described in the “3.3. Analytical sensitivity and reproducibility”.

  1. What is “LoD” in Table 2?

Answer:

 LoD (Limit of Detection)

  1. Does "qPCR" in the title of Table 4 also need the full name?

 Answer:

This has been amended as requested.

quantitative real-time PCR (qPCR)

  1. Why were the results of 4 samples negative in ddPCR? As pointed out in Materials and Methods, are there any criteria negative for ddPCR?

Answer:

Based on Limit of blank (LoB) of ddPCR, we set the cut-off for ddPCR results for values reporting as “undetectable” at less than 2 positive copies/reaction. In ddPCR, 4 samples judged negative were lower than the LoB value (2 positive copies/reaction).

The sensitivity of ddPCR assay is very high, and when the sample does not contain virus or the virus detected is less than 2 copies/reaction, it is judged negative. At the time of testing, ddH2O was used as a negative control.

Discussion

  1. At lines 443~444, the author says this ddPCR could detect most GAstV. GAstV XX strain was used in the present study (L123). What is the basis that other strains can be detected by the ddPCR?

Answer:

This ddPCR method for GAstV detection was established using GAstV XX strain  (GenBank number: MN337323).  The specific primers and probes in this ddPCR assay were designed based on of avian astrovirus Group 1 (GenBank number: MH052598.1).  The specific primers and probes were designed and synthesized based on the ORF2 gene of AstV/SDPY/Goose/1116/17 strain (GenBank number: MH052598.1).

During the isolation and identification of goose astrovirus XX strain (GenBank login number: MN337323), our team has conducted homology analysis and other correlation analysis on the genome, ORF1b and ORF2 genes with representative a large number of strains. The experimental results have been reported.

Genome:

The nucleotide homology of GAstV XX strain and representative strains GD, AstV/SDPY/Goose/1116/17 and AAstV/Goose/CHN/2017/SD01 causing gout in goslings was 98.1%, 98.7% and 98.7%, respectively.

The gene of ORF1b、ORF2:

The nucleotide homology of ORF1b and ORF2 genes of GAstV XX strain with GD, AstV/SDPY/ Goose/1116/17 and AAstV/Goose/CHN/2017/SD01, which were representative strains causing gout in goslings, was 98.6% ~ 99.4%, The homology of amino acid ranged from 98.6% to 99.8%.

Phylogenetic analysis results showed that GAstV XX was in the same evolutionary clade as GD, AstV/SDPY/Goose/1116/17, AAstV/Goose/CHN/2017/SD01. They all belong to avian astrovirus Group 1 [1]. It can be seen that the homogeneity of AstV/SDPY/Goose/1116/17 strain was higher than that of the former viral strain causing the wind pain in goslings. AstV/SDPY/Goose/1116/17 strain is a representative strain of avian astrovirus Group 1. The designed primers and probes can detect most GAstV of avian astrovirus Group 1.

[1] Qianyue Jin, Yonggang Guo, Junpeng Li, et al. Isolation, identification and genetic characterization of goose astrovirus XX strain[J]. Journal of Henan Agricultural Sciences.2021,50(6):134-141. doi:10.15933/ j.cnki.1004-3268.2021.06.016

In short, the isolated GAstV XX strain (GenBank number: MN337323) was sequenced and its genetic characteristics were analyzed. Phylogenetic analysis showed that GAstV XX strain was in the same evolutionary branch as the current circulating strain causing gout in goslings, belonging to group 1 of avian astroviruses, and had a distant evolutionary relationship with duck, Turkey and chicken astroviruses, with great differences. The amino acid sequence analysis of ORF2 encoded protein of different geese strains showed that there was little difference among the circulating strains, and only some mutations of amino acid sites existed.

 Our team is collecting the disease materials of avian astroviruses (duck, chicken, turkey) and will supplement the astroviruses from other birds in future experiments.

  1. Were true positive or true negative of 36 clinical samples confirmed by viral isolation test? 

Answer:

Based on Limit of blank (LoB) of ddPCR, we set the cut-off for ddPCR results for values reporting as “undetectable” at less than 2 positive copies/reaction. In ddPCR, 4 samples judged negative were lower than the LoB value (2 positive copies/reaction).

The Limit of blank (LoB) of ddPCR was added and the results were obtained.

① Limit of blank (LoB) of ddPCR
Using two different batches of reagents, 4 negative samples (blank) were repeated for 3 consecutive days to obtain the value of LoB.

Cp=1.645*[1-(4B-4K)-1] -1 

LoB =M + CpSD

Cp represents the multiplier of the 95th percentile of the normal distribution (α=0.05). B is the number of blank samples during the actual test. K is the number of blank samples. M is mean of blank. SD is standard deviation of blank.

To establish the LoB, we analyzed 4 blank samples with two different batches of reagents for 3 consecutive days. The LoB was estimated nonparametrically as the 95th percentile of the measurements. The LoB of reagent batch 1 was 1.78, and the LoB of reagent batch 2 was 1.81. Both of these were rounded to the whole number 2. That is, the LoB of this method was 2. The LoB of the capsid protein gene (ORF2) gene was calculated to be 2 copies/reaction. We further evaluated using 4 known negative samples and confirmed that the highest copy/reaction number was 2. Therefore, based on this, we set the cut-off for ddPCR results for values reporting as “undetectable” at less than 2 positive copies/reaction.

LoB (Limit of Blank) is an important parameter to judge the detection ability of ddPCR technology. For samples that do not contain analytes (blank samples), LoB is the highest detection result that can be observed. In other words, the maximum number of false positives. The LoB is necessary to determine whether the reaction is positive or negative, and to calculate the LoD (Limit of Detection). In ddPCR, due to the influence of various factors such as reagent components, experimental environment, instruments, etc., even if the sample does not contain the substance to be tested, sometimes the result may not be 0 copy, causing false positives, so it is necessary to define these results that do not have clinical significance, which is LoB. That is, the highest measurement that can be observed in a blank sample.

LoB means that the vast majority (95%) of the zero concentration samples tested are negative and cannot be false positive.

For example, the LoB is 30 copies/mL, although the test result is 10 copies/mL, it is understood that the sample does not contain the substance to be measured, or the concentration is too low to be detected at all. Multiple blank samples can be tested for multiple days and multiple times, and all tests can be statistically analyzed with 95% confidence. In simple terms, LoB defines the standard for 0 concentration samples.

The following results were obtained by supplementing the LoB experiment results. The LoB was calculated to be 2 copies/reaction. We further evaluated using 4 known negative samples and confirmed that the highest copy/reaction number was 2. Therefore, based on this, we set the cut-off for ddPCR results for values reporting as “undetectable” at less than 2 positive copies/reaction.

In summary, false positive results would not be found in clinical samples using the established ddPCR detection method. The identification of LoB can exclude the possibility of false positive in the ddPCR method.

② In the study, the detection limit of ddPCR assay was 10 copies/rection, the virus detection limit of the qPCR assay was 280 copies/µL, which was 28 times higher than that of qPCR detection.The detection rate of ddPCR was indeed higher than that of qPCR in clinical tests with low virus content.

③ The 11 qPCR-negative suspected samples were detected by conventional PCR (amplification with primers of ORF2), and no target bands were amplified.

④ All samples identified positive by qPCR were confirmed by ddPCR.

⑤ PCR amplification products tested positive using ddPCR were encapsulated in oil droplets and cannot be recovered or sequenced.

  1. Is it possible to compare the ddPCR results (copy number) and the qRT-PCR results (copy number) for each of the 36 specimens using a distribution chart or scatter diagram? Because the positive/negative results in Table 5 are not enough to correctly understand the usefulness of ddPCR. Similarly, from the information in Table 5 alone, it is difficult to know whether it can be said that there were no false positives in ddPCR. 

Answer:

LoB (Limit of Blank) is an important parameter to judge the detection ability of ddPCR technology. For samples that do not contain analytes (blank samples), LoB is the highest detection result that can be observed. In other words, the maximum number of false positives. The LoB is necessary to determine whether the reaction is positive or negative, and to calculate the LoD (Limit of Detection). In ddPCR, due to the influence of various factors such as reagent components, experimental environment, instruments, etc., even if the sample does not contain the substance to be tested, sometimes the result may not be 0 copy, causing false positives, so it is necessary to define these results that do not have clinical significance, which is LoB. That is, the highest measurement that can be observed in a blank sample.

LoB means that the vast majority (95%) of the zero concentration samples tested are negative and cannot be false positive.

Therefore, the purpose of LoB determination in ddPCR is to avoid false positive results.

  1. Comments on the Quality of English Language

There were some sentences that appeared to have incorrect grammar.

Answer:

The paper has been revised. Certificate as follows,

Round 2

Reviewer 1 Report

Comments and Suggestions for Authors

The authors have addressed all comments

Author Response

Dear Editor,

     The manuscript has been revised by a professional retouching company, and the certificate is as follows.

     At the same time, I also made several modifications, focusing on Table 2. Blue font displayed.

      Line339: (Not detected) was deleted.

      Line355: (Limit of Detection) was deleted.

      Line357:  ND = Not detected; NTC = Negative control, ddH2O; LoD = Limit of Detection.    This sentence was added.

Editing Certificate
